# Sleep Quality in Patients Undergoing Transcatheter Aortic Valve Implantation (TAVI)

**DOI:** 10.3390/ijerph18168889

**Published:** 2021-08-23

**Authors:** Giulia Lorenzoni, Danila Azzolina, Chiara Fraccaro, Caterina Zoccarato, Clara Minto, Sabino Iliceto, Dario Gregori, Giuseppe Tarantini

**Affiliations:** 1Unit of Biostatistics, Epidemiology and Public Health, Department of Cardiac, Thoracic, Vascular Sciences and Public Health, University of Padova, 35131 Padova, Italy; giulia.lorenzoni@unipd.it (G.L.); danila.azzolina@unife.it (D.A.); caterina.zoccarato.1@studenti.unipd.it (C.Z.); mintoclara@gmail.com (C.M.); 2Department of Medical Sciences, University of Ferrara, 44121 Ferrara, Italy; 3Interventional Cardiology Unit, Department of Cardiac, Thoracic, Vascular Sciences and Public Health, University of Padova, 35128 Padova, Italy; chiara.fraccaro@aopd.veneto.it (C.F.); giuseppe.tarantini.1@unipd.it (G.T.); 4Cardiology Unit, Department of Cardiac, Thoracic, Vascular Sciences and Public Health, University of Padova, 35128 Padova, Italy; sabino.iliceto@unipd.it

**Keywords:** sleep quality, Transcatheter Aortic Valve Implantation (TAVI), quality of life

## Abstract

The present study aimed to analyze sleep quality and quality of Life (QoL) in patients undergoing Transcatheter Aortic Valve Implantation (TAVI). It was conducted at the Interventional Cardiology Unit of the Department of Cardiac, Thoracic, Vascular Sciences and Public Health of the University of Padova on 27 adult patients who underwent TAVI via the transfemoral approach. Patients completed two validated instruments, i.e., the Pittsburgh Sleep Quality Index (PSQI) and the EuroQoL (EQ-5D-5L), on the day of discharge and one month after the hospital discharge. Twenty-seven patients were enrolled with a severe aortic stenosis diagnosis, treated with transfemoral TAVI procedure. The study population included seventeen poor sleepers and ten good sleepers with a median age of 81.92 years overall. The global PSQI evaluation revealed a small significant improvement at follow-up (*p*-value 0.007). Small positive changes were detected in the Self-care and Usual activity domains of the EQ-5D-5L and the EQ-VAS. No correlation was detected between EQ-5D-5L and sleep quality. The present study confirms the importance of sleep quality monitoring in patients who undergo TAVI procedure for aortic stenosis treatment.

## 1. Introduction

The wellbeing of patients is not only related to medical interventions, surgical procedures, or pharmacological treatments, but it is also associated with the quality of daily activities such as eating, physical activity, or sleep [1]. Sleep plays a vital role in maintaining good health status [2]. Beyond scientific theories on the role of sleep on human homeostasis, this function’s importance is demonstrated by the health problems observed in acute or chronic sleep deprivation. These include negative consequences [3] on cardiovascular and immune systems, respiratory capacity, metabolic homeostasis, and cognitive ability. 

The relationship between sleep patterns and cardiovascular health is relevant from the public health perspective since cardiovascular diseases represent a leading cause of death. Large cohort studies have shown that sleep problems, e.g., short or long sleep duration, sleep arousal, insomnia, are associated with cardiovascular diseases onset and cardiovascular mortality [4,5]. Furthermore, it has been shown that patients with cardiovascular diseases are more likely to present with sleep problems, which are, in turn, associated with a poor prognosis [6]. Most of the studies investigating sleep problems in patients with cardiovascular diseases focus on heart failure and ischemic heart diseases.

Conversely, data on sleep quality in patients with valvular disease are scarce, despite the fact valvular diseases represent a relevant public health problem. With the increase of life expectancy [7], aortic stenosis, one of the most common valvular diseases, has become a widespread pathological condition with a prevalence of 5% at 65 years [8]. Without treatment, these patients have a poor QoL and a severe prognosis. For many years, surgical aortic valve replacement had represented the standard of care for severe aortic stenosis. Recently, Transcatheter Aortic Valve Implantation (TAVI) has emerged as an alternative to the traditional surgical approach in selected patients [9], allowing for replacing the stenotic valve using a percutaneous approach instead of invasive surgery. 

Patients with aortic stenosis have been identified to be at risk of sleep disturbances [10]. In addition to the fact that aortic stenosis per se is a risk factor for sleep problems, the treatment for aortic stenosis may be associated with sleep disturbances. In the immediate postoperative phase, sleep difficulties have been reported in both patients treated with surgical approach and TAVI patients [11]. It is crucial to diagnose and treat sleep disturbances that arise during the immediate postoperative phase because they may result in chronic sleep disorders, impaired Quality of Life (QoL), and incomplete rehabilitation [12]. Furthermore, it has been demonstrated that sleep disorders that originate during acute illness can persist several months after discharge [13]. 

In patients undergoing cardiac procedures, sleep quality seems to be related to QoL’s general concept [14,15,16]. Sleep quality disturbances are generally reported preoperatively and during the hospital stay, but in the long-term (after hospital discharge), both QoL and sleep quality improvement may be observed [17]. However, such studies have focused on heart surgery procedures. When TAVI patients were considered, studies have focused on the immediate postoperative step, while evidence on sleep monitoring after the discharge from the TAVI procedure is lacking. The present study aims to fill this gap by analyzing patients’ perceived sleep quality and QoL at discharge after the TAVI procedure and one month after discharge. 

## 2. Materials and Methods

The study was conducted in the Interventional Cardiology Unit of the Department of Cardiac, Thoracic, Vascular Sciences and Public Health of the University of Padova (Padova, Italy) between May and June 2017. All subjects who underwent TAVI during the period of data collection were screened for possible inclusion. Only adult patients diagnosed with aortic stenosis undergoing a TAVI procedure via the transfemoral approach were enrolled. Criteria for eligibility were the absence of major cognitive impairment and the native Italian language. Subjects with severe clinical comorbidities (terminal cancer, renal or liver failure) or treated with transapical approach were excluded. Informed consent was given to all subjects enrolled after the explanation of the study’s objectives and design. Questionnaires were administered at discharge and one month after hospital discharge. Study design, instruments of data collection, and data analysis are in full respect of ethical guidelines established by the American Psychological Association and the Italian Association of Psychology [18]. 

### 2.1. Instruments of Data Collection

Patients completed two validated instruments: the Pittsburgh Sleep Quality Index (PSQI) and the EuroQol (EQ-5D-5L). The average time to complete both questionnaires was 30 min. Additional information was collected, including age, gender, comorbidities, procedure date, and pharmacological treatment. Data were recorded using REDCap (Research Electronic Data Capture), a web-based tool for data collection [19].

### 2.2. The Pittsburgh Sleep Quality Index 

The PSQI is a self-administered questionnaire assessing sleep quality. PSQI includes nineteen individual items generating 7 component scores: perceived sleep quality, sleep latency, sleep duration, habitual sleep efficiency, sleep disturbances, use of sleeping medication, and daytime dysfunction [20]. Each component’s score ranges from 0 to 3 points, where “0” indicates no difficulty and “3” indicates severe difficulty. Results of PSQI can be either evaluated for each component or by calculating the global score. The latter corresponds to the sum of the seven component scores, ranging from 0 to 21, where lower scores denote a better sleep quality. A categorization of the PSQI has been proposed, defining “good sleepers” subjects with a global PSQI score less than or equal to 5 points and “poor sleepers” subjects with a score higher than 5 points [20]. Moreover, it has been shown that the proposed cut-off, applied to the Italian population, has good performance [21].

### 2.3. The EuroQol Five Dimensions Instrument

The EQ-5D-5L was employed to evaluate patients’ perceived QoL [22]. It includes a visual analog scale (EQ-VAS) and five different dimensions: mobility, self-care, usual activities, pain or discomfort, and anxiety or depression. The subject can select one of the five response levels for each dimension, reporting no problems, slight problems, moderate problems, severe problems, and extreme problems. The EQ-VAS records the global health rating on a 0–100 scale, where “100” is the “best health you can imagine.” 

### 2.4. Power Analysis

The study was designed to enroll 25 subjects. The outcome was the difference in the PSQI scale between 1-month follow-up and baseline. The expected difference was assumed to be normally distributed with a standard deviation of 2 PSQI scores. If the true difference in the mean response would be of 2 points of PSQI scores, we could reject the null hypothesis with a power of 0.996. The Type I error probability was 0.05. Power remained above 0.80 for a standard deviation of up to 3.4.

### 2.5. Statistical Analysis

Descriptive statistics were reported using absolute number and percentage for categorical variables and median and interquartile range (I–III quartile) for continuous data. The Wilcoxon-Kruskal–Wallis and the Chi-squared tests were done to compare continuous and categorical variables distribution between good sleepers (PSQI ≤ 5) and poor sleepers (PSQI > 5). 

A mixed-effects model for longitudinal analysis was applied to evaluate changes in both questionnaires’ scores from baseline to follow-up, defined as the number of days between enrolment and the follow-up examination. Changes in sleep quality were estimated through the analysis of the seven PSQI components (sleep quality, sleep latency, sleep duration, habitual sleep efficiency, sleep disturbances, use of sleeping medication, and daytime dysfunction) and the global PSQI score. In the same way, we analyzed the perceived QoL by evaluating the five dimensions of the EQ-5D-5L questionnaire (mobility, self-care, usual activities, pain/discomfort, and anxiety/depression) and the EQ-VAS. For longitudinal analyses, *p*-values were adjusted to account for multiplicity using the approach proposed by Benjamini and Hochberg [23]. Furthermore, the correlation between PSQI and EQ-5D-5L was evaluated using a Multiple Marginal GEE Model [24,25].

Statistical analysis was performed using the R System version 3.3.2 [26] and mmmgee [27], gee [28], multcomp [29], and rms [30] packages.

## 3. Results

Twenty-seven patients were enrolled with a severe aortic stenosis diagnosis, treated with transfemoral TAVI procedure during the study period. As shown in Table 1, the population included seventeen poor sleepers and ten good sleepers with a median age of 81.92 years overall. The prevalence of comorbidities was high, especially of cardiovascular, metabolic, and genitourinary diseases (89%, 81%, and 41%, respectively). No significant differences were detected between poor and good sleepers regarding comorbidities prevalence, except for cardiovascular diseases that were found to be significantly more prevalent among poor sleepers. No significant differences were detected in the distribution of pharmacological treatments (Table 2).

Results of sleep quality evaluation are reported in Table 3. A significant difference between baseline and follow-up was detected for perceived sleep quality and daytime dysfunction. No other changes were detected, and the single components’ scores remained stable between baseline and follow-up. The global PSQI evaluation revealed a slight significant improvement. Table 4 reports the QoL, measured with the EQ-5D-5L. Small positive changes were detected in Self-care and Usual activity domains, as well as in the EQ-VAS. 

The correlation among the outcomes (PSQI and EQ-5D-5L) evaluated using GEE was not significant (−0.17, 95% C.I. −0.42; 0.09).

## 4. Discussion

The present study evaluated sleep quality in patients undergoing TAVI procedure for the treatment of aortic stenosis. Our findings showed an overall poor sleep quality, both at baseline and follow-up evaluations, and only a slight improvement after valve replacement. For what concerns QoL, it has been shown an improvement of self-care ability and usual activity performance, together with the EQ-VAS. 

Several hypotheses have been proposed to understand the role of sleep habits in wellbeing maintenance. Researchers in the field agree that sleep is essential for energy conservation [31]. For example, differences in sleep duration between mammals are strictly associated with the diet, and metabolic request, e.g., omnivores and carnivores sleep longer than herbivores. 

Sleep disturbance among patients with clinical and subclinical cardiovascular disorders has been widely explored in the scientific literature. In a recent meta-analysis on the effects of sleep duration, Cappuccio et al. reported that both short and long sleep duration could deteriorate patients’ clinical condition, exposing them to a higher risk of death for coronary heart disease and stroke [32]. This trend seems to be even more pronounced in female and elderly subjects sleeping less than four hours per night [33]. 

Although a growing body of literature shows an association between sleep problems and heart disorders, the literature on TAVI patients is lacking. A cohort study comparing octogenarians patients treated with TAVI or traditional surgical approach reported a high incidence of early postoperative insomnia in subjects who underwent TAVI [11]. The high prevalence of sleep disturbances among TAVI patients, which is consistent with the present work results, is probably the consequence of heterogeneous risk factors related to the surgical procedure, individual characteristics, and pharmacological treatments. Even though it is difficult to understand each variable’s weight in determining the poor sleep quality, we might hypothesize that comorbidities, age, and medications are the most relevant. Generally, patients treated with TAVI are older than those who undergo surgical aortic valve replacement. Furthermore, they are often described as fragile patients suffering from several comorbidities requiring polytherapy. Each one of these factors is associated with a higher risk of sleep disturbances. The aging process itself causes a general increase in the proportion of light sleep and a progressive decrease in the proportion of deep-sleep states, including the REM phase. These changes led to a substantial reduction of restorative sleep with increased transitory sleep [34]. Respiratory disorders, such as chronic obstructive pulmonary disease (COPD), and obesity often cause sleep apnea episodes, affecting sleep quality. In addition to that, mental illness, depression, dementia, and delirium play a relevant role in sleep disturbances [35]. Not least, several pharmacological treatments may disrupt sleep patterns. Beta-blockers, generally prescribed in hypertension, can inhibit melatonin secretion, causing frequent awakenings during the night [36]. ACE inhibitors may increase the amount of circulating bradykinin and affect potassium levels. Patients can suffer from sudden cough and leg cramps during sleep [37]. The effects of statins are still debated, even though a meta-analysis reported no significant adverse impact of such medications on sleep duration and efficiency [38]. 

Nowadays, medical research agrees on the relevance of sleep habits on human health and the need for their monitor, especially in critically ill and at-risk patients and in subjects unable to communicate (such as children or patients with dementia). Until now, objective evaluation of sleep patterns has been made using polysomnography, a diagnostic test used to record brain waves, oxygen level, heart rate, eye, and leg movement during sleep. Biomedical science is now interested in using wearable devices capable of evaluating sleep quality in everyday life. Such devices have also been proposed for TAVI patients [39]. Home monitoring devices may offer a more realistic picture of sleep patterns, especially in chronic diseases that require long-term or long-life management. 

For what concerns the study limitations, the small sample size represents the main one. Although the power analysis was supporting our study regarding sample size, the number of patients enrolled was small, thus limiting the robust generalizability of results. Furthermore, a preoperative assessment was not done, so that we cannot make inferences about acute changes in sleep quality in the immediate postoperative period. 

## 5. Conclusions

The present study confirms the importance of sleep quality monitoring in fragile patients, such as subjects treated for aortic stenosis with the TAVI procedure. More efforts should be put forward to investigate sleep patterns in such patients outside the hospital setting to assess the long-term impact of the procedure on sleep quality. 

## Figures and Tables

**Table 1 ijerph-18-08889-t001:** Characteristics of the study population. Continuous data are reported as median (I, III quartiles), categorical data are reported as absolute numbers (percentage). Good sleepers are subjects with a Global PSQI Score equal to or less than 5 points, poor sleepers are subjects with a Global PSQI Score higher than 5 points.

	Good Sleepers (*n* = 10)	Poor Sleepers (*n* = 17)	Combined (*n* = 27)	*p*-Value
Females	5 (50)	12 (71)	17 (63)	0.29
Age	81.69 (79.91, 84.52)	82.54 (76.35, 84.24)	81.92 (78.27, 84.53)	0.69
Comorbidities				
Cardiovascular diseases	7 (70)	17 (100)	24 (89)	0.02
Metabolic diseases	8 (80)	14 (82)	22 (81)	0.88
Respiratory diseases	2 (20)	3 (18)	5 (19)	0.88
Musculoskeletal diseases	2 (20)	4 (24)	6 (22)	0.83
Gastrointestinal diseases	3 (30)	7 (41)	10 (37)	0.56
Genito-urinary diseases	5 (50)	6 (35)	11 (41)	0.45
Psychiatric diseases	0 (0)	2 (12)	2 (7)	0.26
Neurological diseases	0 (0)	1 (6)	1 (4)	0.43
Familiarity for any diseases	4 (40)	4 (24)	8 (30)	0.37
Others	7 (70)	7 (41)	14 (52)	0.15

**Table 2 ijerph-18-08889-t002:** Pharmacological therapy. Data are reported as absolute numbers (percentage). Good sleepers are subjects with a Global PSQI Score equal to or less than 5 points, Poor sleepers are subjects with a Global PSQI Score higher than 5 points.

	Good Sleepers (*n* = 10)	Poor Sleepers (*n* = 17)	Combined (*n* = 27)	*p*-Value
Antiarrhythmic drugs				
None	9 (90)	14 (82)	23 (85)	0.59
Amiodarone	1 (10)	1 (6)	2 (7)	0.69
Digitale	0 (0)	1 (6)	1 (4)	0.43
Isoptin	0 (0)	1 (6)	1 (4)	0.43
Antiplatelet drugs				
None	3 (30)	3 (18)	6 (22)	0.46
Acetylsalicylic acid	3 (30)	10 (59)	13 (48)	0.15
Others	4 (40)	7 (41)	11 (41)	0.95
Antihypertensives and vasodilators				
None	7 (70)	6 (35)	13 (48)	0.08
Amlodipine	0 (0)	2 (12)	2 (7)	0.26
Olmesartan	0 (0)	1 (6)	1 (4)	0.43
Ibesartan	0 (0)	1 (6)	1 (4)	0.43
Losartan	0 (0)	1 (6)	1 (4)	0.43
Nitroglycerin	1 (10)	0 (0)	1 (4)	0.18
Valsartan	1 (10)	0 (0)	1 (4)	0.18
Others	1 (10)	8 (47)	9 (33)	0.05
Cholesterol-lowering drugs and statins				
None	7 (70)	6 (35)	13 (48)	0.08
Atorvastatin	1 (10)	7 (41)	8 (30)	0.09
Pravastatin	0 (0)	1 (6)	1 (4)	0.43
Rosuvastatin	0 (0)	1 (6)	1 (4)	0.43
Simvastatin	1 (10)	1 (6)	2 (7)	0.69
Others	1 (10)	2 (12)	3 (11)	0.89
Ace-inhibitors				
None	7 (70)	10 (59)	17 (63)	0.56
Enalapril	0 (0)	2 (12)	2 (7)	0.26
Ramipril	3 (30)	5 (29)	8 (30)	0.97
Diuretics				
None	5 (50)	8 (47)	13 (48)	0.88
Furosemide	5 (50)	6 (35)	11 (41)	0.45
Potassium canrenoate	3 (30)	4 (24)	7 (26)	0.71
Others	0 (0)	5 (29)	5 (19)	0.06
Beta-blockers				
None	6 (60)	6 (35)	12 (44)	0.21
Bisoprolol	4 (40)	7 (41)	11 (41)	0.95
Metoprolol	0 (0)	3 (18)	3 (11)	0.16
Others	0 (0)	1 (6)	1 (4)	0.43
Other drugs				
None	0 (0)	1 (6)	1 (4)	0.43
Anticoagulants	1 (10)	2 (12)	3 (11)	0.89
Insulin	2 (20)	2 (12)	4 (15)	0.56
Hypoglyemic agents	2 (20)	4 (24)	6 (22)	0.83
Gastroprotectants	6 (60)	14 (82)	20 (74)	0.20
Sleep-inducing drugs	2 (20)	4 (24)	6 (22)	0.83
Others	8 (80)	12 (71)	20 (74)	0.59

**Table 3 ijerph-18-08889-t003:** Single components and global PSQI scores at baseline and follow-up and their estimated change using longitudinal models. Data are reported as the median and interquartile range at baseline and follow-up. Results of the longitudinal model are reported as estimated effect (EE), standard error (SE), and *p*-value.

	Baseline	Follow-Up	*p*-Value ^1^	EE	SE	*p*-Value ^2^
Perceived sleep quality	1.00 (1.00, 1.00)	1.00 (0.00, 1.00)	0.02	−0.26	0.10	0.02
Sleep latency	1.00 (0.00, 2.00)	1.00 (0.00, 2.00)	0.35	0.08	0.05	0.17
Sleep duration	1.00 (0.00, 2.00)	1.00 (0.00, 2.00)	0.35	−0.11	0.10	0.29
Habitual sleep efficiency	1.00 (0.00, 2.50)	1.00 (0.00, 2.00)	0.71	−0.07	0.20	0.71
Step disturbance	1.00 (1.00, 1.00)	1.00 (0.00, 1.00)	1.00	−0.04	0.04	0.33
Use of sleep medication	0.00 (0.00, 2.00)	1.00 (0.00, 1.00)	0.37	−0.19	0.13	0.17
Daytime dysfunction	1.00 (0.50, 1.00)	1.00 (0.00, 1.00)	0.005	−0.37	0.11	0.002
**Global PSQI Score**	7.00 (4.00, 10.00)	6.00 (3.50, 7.50)	0.007	−1.07	0.37	0.008

^1^*p*-value for the difference between baseline and follow-up values ^2^*p*-value for the estimated change of the longitudinal model.

**Table 4 ijerph-18-08889-t004:** EQ-5D-5L single components evaluation and EQ-VAS scores at baseline and follow-up and their estimated change using longitudinal models. Data are reported as the median and interquartile range at baseline and follow-up. Results of the longitudinal model are reported as estimated effect (EE), standard error (SE), and *p*-value.

	Baseline	Follow-Up	*p*-Value ^1^	EE	SE	*p*-Value ^2^
Mobility	2.00 (1.00, 3.00)	2.00 (1.00, 2.00)	0.07	−0.222	0.111	0.06
Self-care	2.00 (2.00, 2.00)	2.00 (1.50, 2.00)	0.05	−0.222	0.097	0.03
Usual activity	2.00 (2.00, 2.00)	2.00 (1.00, 2.00)	0.03	−0.296	0.117	0.02
Pain	1.00 (1.00, 2.00)	1.00 (1.00, 1.50)	0.06	−0.259	0.126	0.05
Anxiety	1.00 (1.00, 1.00)	1.00 (2.00, 1.00)	0.53	0.074	0.106	0.49
EQ-VAS	60.00 (50.00, 80.00)	70.00 (60.00, 80.00)	<0.001	6.111	1.370	<0.001

^1^*p*-value for the difference between baseline and follow-up values ^2^*p*-value for the estimated change of the longitudinal model.

## Data Availability

The data presented in this study are available on request from the corresponding author.

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
