# Peer review of "Sleep Quality in Patients Undergoing Transcatheter Aortic Valve Implantation (TAVI)"

_ijerph, 2021, doi:10.3390/ijerph18168889_

Round 1

Reviewer 1 Report

This article entitled " Sleep Quality in Patients Undergoing Transcatheter Aortic 2 Valve Implantation (TAVI)" is well-written and conducted in scientific manner. The discussion and conclusion seems to be reasonable and acceptable. 

But, I have one question. Why sleep quality is important for patients underwent TAVI procedure? (not for general population, as you described in page 1, line 36 to 41). 

1. Did procedural results affect the quality of sleep? 

2. Or did the quality of sleep affect follow-up result?

This interesting observation for quality of sleep after TAVI would be better to be extended to longer duration (at least three or six months) 

Author Response

This article entitled " Sleep Quality in Patients Undergoing Transcatheter Aortic 2 Valve Implantation (TAVI)" is well-written and conducted in scientific manner. The discussion and conclusion seems to be reasonable and acceptable. 

  1. But, I have one question. Why sleep quality is important for patients underwent TAVI procedure? (not for general population, as you described in page 1, line 36 to 41). 

We would like to thank the reviewer for the comment. It has been better specified the relationship between sleep problems and cardiovascular diseases in the introduction.

  1. Did procedural results affect the quality of sleep? Or did the quality of sleep affect follow-up result?

We would like to thank the reviewer for the question that allows us to clarify the concept reported in the paper. The study was aimed at assessing the quality of sleep immediately after and one-month after discharge from the TAVI procedure. Interestingly, the analysis of the PSQI score revealed a small significant improvement at follow-up, even though the sleep quality remained poor. So, we may hypothesize that the TAVI procedure, in the short-term, results in slightly better quality of sleep. However, we cannot make inference about acute changes in quality of sleep since the preoperative assessment was not done. Such concepts have been reported in the discussion section.

  1. This interesting observation for quality of sleep after TAVI would be better to be extended to longer duration (at least three or six months) 

We would like to thank the reviewer for the comment that has been included in the Conclusions section.

Reviewer 2 Report

The Authors evaluated 27 patients undergoing Transcatheter Aortic Valve Implantation (TAVI) thorough a cross-sectional study analyzing the sleep quality (by Pittsburgh Sleep 20 Quality Index - PSQI) and quality of life (by EuroQoL - EQ-5D-5L) the day of discharge and one month later. They found a small significant improvement at follow-up global of PSQI and of Self-care and Usual activity domains of the EQ-5D-5L, as well as in the final EQ-VAS, without significant correlation between EQ-5D-5L and sleep quality.

The topic of this study is interesting and timely, as it contributes to the current discussion about the relationship between sleep and cardiovascular risks and clinical outcome. Even if, generally speaking, the study had been well conducted and explained, some issues reduce its strength and clinical significance.

In brief:

  • the small sample size, as clearly disclosed by Authors in study limitations
  • the lacking of a paired control group
  • considering the small sample size, the presence of specific sleep disorders (central and obstructive sleep apnea, restless leg syndrome and periodic limb movement disorders, insomnia) as well as psychological functions (mood and anxiety) have not been evaluated by history and instrumental devices (polysomnography and actigraphy)
  • a preoperative evaluation of sleep and quality of life has not been performed
  • insufficient scientific basis are provided to support the research question and results discussion: the global sleep quality and quality of life have been evaluated without regards to specific sleep disorders as well as psychological functions given the well-known association of these disorders and cardiovascular diseases

Unfortunately, the paper, as it is, needs relevant changes in methods and, consequently, a further full revision before to be published on IJERPH.

Author Response

In brief:

  • the small sample size, as clearly disclosed by Authors in study limitations

That of the small sample size is a study limitation that has been already declared in the discussion section. Anyway, even small, a sample of 25 patients allowed for the detection of a change of 2 points in the PSQI score with a power of 0.99.

  • the lacking of a paired control group

the aim of the study was to evaluate sleep quality immediately after and one-month after TAVI not to compare TAVI patients’ sleep quality with that of subjects without aortic stenosis (controls), so there is no rationale to enroll a control group.

  • considering the small sample size, the presence of specific sleep disorders (central and obstructive sleep apnea, restless leg syndrome and periodic limb movement disorders, insomnia) as well as psychological functions (mood and anxiety) have not been evaluated by history and instrumental devices (polysomnography and actigraphy)

At baseline, information about drug therapy was collected including sleep inducing drugs. The proportion of patients taking sleep inducing drugs was low (6 out of 27, without significant differences in the proportion of subjects taking these drugs between good and poor sleepers).

  • a preoperative evaluation of sleep and quality of life has not been performed

This is a study limitation that has been included in the Discussion section.

  • insufficient scientific basis are provided to support the research question and results discussion: the global sleep quality and quality of life have been evaluated without regards to specific sleep disorders as well as psychological functions given the well-known association of these disorders and cardiovascular diseases

The impact of perceived sleep quality is of growing interest in the field of cardiovascular research, and it has been put in relationship with cardiovascular morbidity and mortality. Several instruments have been used to assess sleep quality in cardiovascular research, including the PSQI (1–3). So, not only sleep disorders are a matter of concern for cardiovascular health, but also perceived sleep quality. The introduction section has been improved to strength the study rationale.

  1. Chen H-C, Su T-P, Chou P. A nine-year follow-up study of sleep patterns and mortality in community-dwelling older adults in Taiwan. Sleep. 2013;36(8):1187–98.
  2. Sharma M, Sawhney JPS, Panda S. Sleep quality and duration–Potentially modifiable risk factors for Coronary Artery Disease? indian heart journal. 2014;66(6):565–8.
  3. Wang Q, Cheng HY, Lo SW-S, Li XM, Wong EM-L, Sit JW-H. Relationship between sleep quality and cardiovascular disease risk in Chinese post-menopausal women. BMC women’s health. 2017;17(1):1–7.

Unfortunately, the paper, as it is, needs relevant changes in methods and, consequently, a further full revision before to be published on IJERPH.

We would like to thank the reviewer for his/her useful comments that allow us to improve the manuscript. A point-by-point response has been provided and manuscript has been amended according to the comments.

Round 2

Reviewer 1 Report

Thank you for your comprehensive revision.

I have one comment

In discussion section, please add the discussion of relationship between sleep quality and diuretics.